# Prevalence and Rate of Resolution of Left Atrial Thrombus in Patients with Non-Valvular Atrial Fibrillation: A Two-Center Retrospective Real-World Study

**DOI:** 10.3390/jcm11061520

**Published:** 2022-03-10

**Authors:** Pompilio Faggiano, Elisabetta Dinatolo, Antonella Moreo, Benedetta De Chiara, Marco Sbolli, Francesco Musca, Antonio Curnis, Oriana Belli, Cristina Giannattasio, Cesare Tomasi, Marco Metra, Gloria Santangelo

**Affiliations:** 1Fondazione Poliambulanza, 25124 Brescia, Italy; elisabetta.dinatolo@poliambulanza.it; 2Division of Cardiology, Department of Medical and Surgical Specialties, Radiological Sciences, and Public Health, University and Civil Hospital of Brescia, 25123 Brescia, Italy; antonio.curnis@libero.it (A.C.); marco.metra@libero.it (M.M.); 3A. De Gasperis Cardio Center, ASST Grande Ospedale Metropolitano Niguarda and Department Scienze della Salute, Bicocca University, 20126 Milan, Italy; antonella.moreo@gmail.com (A.M.); benedetta.dechiara@ospedaleniguarda.it (B.D.C.); sbollimarco@gmail.com (M.S.); francesco.musca@ospedaleniguarda.it (F.M.); oriana.belli@ospedaleniguarda.it (O.B.); cristina.giannattasio@ospedaleniguarda.it (C.G.); 4Section of Occupational Medicine and Industrial Hygiene, University of Brescia, 25121 Brescia, Italy; cesare.tomasi@live.com; 5Division of Cardiology, Department of Health Sciences, San Paolo Hospital, University of Milan, 20122 Milano, Italy; gloriasantangelo@hotmail.it

**Keywords:** left atrial appendage thrombus, atrial fibrillation, vitamin K antagonist, direct oral anticoagulants, heparin

## Abstract

Background and aim. Thromboembolic events due to left atrial appendage (LAA) thrombosis are the main complication of non-valvular atrial fibrillation (NVAF). Although anticoagulants are effective in patients with NVAF, a minimal residual thromboembolic risk persists. Little is known about the prevalence of LAA thrombus and the rate of resolution after the recommended period of anticoagulation therapy, including vitamin K antagonists (VKA), heparin, and non-vitamin K antagonist oral anticoagulants (NOACs). Methods and results. We aimed to study the prevalence of LAA thrombus in an unselected cohort of patients undergoing transesophageal echocardiogram (TEE), and the determinants of LAA thrombus resolution. We retrospectively analyzed 8888 consecutive TEEs performed over five years in two high-volume centers and included all patients with LAA thrombus. A total of 265 patients (3%) had an LAA thrombus. Among these, 97% presented with AF. Fifty-eight percent of patients were on anticoagulants at least three weeks before the diagnosis. After the LAA thrombus diagnosis, VKAs were prescribed in 52%, heparin in 18.5%, and NOAC in 27% of patients. Among the 183 patients with repeat TEE, performed at (25–75th) 39 days (21–84), 67% showed resolution of the LAA thrombus. Although the rate of thrombus resolution was higher in patients treated with NOACs (NOACs 71%, VKA 66%, Heparin 60%) the difference between anticoagulants was statistically non-significant (VKA, OR 0.9, *p* = 0.83; NOAC, OR 1.23, *p* = 0.42; heparin, OR 0.69, *p* = 0.35). Thus, NOACs were demonstrated to be at least as effective as other anticoagulants in the rate of LAA thrombus resolution. Upon multivariate-adjusted analysis, higher LAA emptying velocities were the only predictor of thrombus resolution. In conclusion, the majority of patients were already on anticoagulants. NOACs could be at least as effective as other anticoagulants, yielding an LAA thrombus resolution in two-thirds of patients. This may have clinical relevance, especially in patients undergoing cardioversion or catheter ablation.

## 1. What Is New

In patients with left atrial appendage thrombus, repeated transesophageal echocardiograms may document a persistence of thrombotic apposition, despite anticoagulant therapy. NOACs could be at least as effective as other anticoagulants.

A careful strategy should be implemented in atrial fibrillation patients with left atrial appendage thrombus before elective cardioversion or ablative approach.

## 2. Introduction

Non-valvular atrial fibrillation (NVAF) is associated with a 5-fold increased risk of thromboembolic events compared to the general population [1]. Several studies have reported a 20–25% prevalence of NVAF in patients admitted to hospital after stroke [2], and this percentage results in being higher if the patients undergo a prolonged in-hospital heart rhythm monitoring. Prolonged in-hospital heart rhythm monitoring allows for a higher rate of AF detection. Thromboembolic events due to left atrial appendage (LAA) thrombosis are the most feared complication of NVAF. Thromboembolic risk remains increased even in the early period after electrical cardioversion [3]. Non-vitamin K antagonist oral anticoagulants (NOACs) were demonstrated to be at least non-inferior to vitamin K antagonists (VKA) in preventing thromboembolism in randomized clinical trials (RCTs) [1]. The efficacy of NOACs in patients suitable for electrical cardioversion or with LAA thrombus diagnosis is increasingly supported by scientific evidence, even though patient adherence to this anticoagulant approach remains undetermined [4,5].

After a diagnosis of LAA thrombus by transesophageal echocardiogram (TEE), international guidelines [1] recommend effective anticoagulation for at least three weeks (class I, level of evidence C), and a repeat TEE should be performed to ensure thrombus resolution (class IIa, level of evidence C). However, little is known about the prevalence of LAA thrombus in an unselected population and, particularly, the rate of resolution after-the recommended period of anticoagulation therapy.

The aims of this real-word retrospective and observational study were to establish:the prevalence of LAA thrombus in an unselected cohort of patients undergoing TEE;the prevalence of LAA thrombus despite recommended anticoagulant therapy;the rate of LAA thrombus resolution on repeat TEE after at least three weeks of anticoagulation;the efficacy of different anticoagulant regimens (including VKA, heparin, and NOACs);clinical and echocardiographic determinants of LAA thrombus persistence.

## 3. Methods

We reviewed 8888 consecutive TEE studies performed between January 2014 and December 2018 in two high-volume non-invasive cardiac imaging laboratories, at the Niguarda Hospital in Milan, Italy (Center 1), and the Civil Hospital in Brescia, Italy (Center 2) (Figure 1). TEEs were retrieved from institutional TEE databases. Two independent investigators for each of the two centers reviewed off-line electronically stored echocardiographic images that were collected. In 20 randomly selected and blindly analyzed studies, there was a good intra- and inter-observer agreement.

For LAA thrombus prevalence estimates, the patient’s first thrombus identification has been considered. According to the 2010 European Society of Cardiology Recommendations for echocardiography use in the diagnosis and management of cardiac sources of embolism, thrombi were defined as “echo reflecting masses in the atrial body or in the LAA (often in its apex), distinct from the underlying endocardium, observed in more than one imaging plane, and not related to pectinate muscles” [6].

Despite evidence of an association between sludge and thromboembolism [4], we did not assess LAA sludge or spontaneous echo contrast (SEC) in our study because of the lack of a standardized and univocal echocardiographic definition.

Demographic and clinical characteristics were collected from electronic medical records. According to ESC guidelines [1], atrial fibrillation is defined as paroxysmal if lasting <7 days with self-termination, persistent if lasting >7 days or terminated by cardioversion, and permanent if NVAF is accepted and rhythm control strategy is abandoned. In addition, electronic charts were reviewed for documented anticoagulant therapy.

Echocardiographic variables available were collected from our echocardiogram database. We identified patients with LAA thrombus who had a repeat TEE within one year to obtain estimates of thrombus persistence and/or resolution. In addition, thromboembolic complications such as stroke, transient ischemic attacks, or peripheral embolism were identified from chart review.

The Ethic Committee approved this study, and, due to the retrospective nature of the study, individual patient informed consent was not provided (with agreement of Ethic Committee).

## 4. Statistical Analysis

The continuous variables were submitted to Kolmogorov-Smirnov test, to identify the possible approximation to the normal Gaussian distribution. Continuous variables were shown as median and IQR (25–75th percentile) and compared with Mann–Whitney U test. Categorical variables were presented as frequencies or percentages and compared with the use of the Chi-Square test and the Fisher’s exact test, as appropriate; data were shown as odds ratios (OR) related to the Chi-square test for 2 × 2 tables and 95% confidence intervals (CIs). Finally, a logistic multivariate analysis was performed among clinical and echocardiographic parameters to identify predictors of thrombus resolution (see table below). IBM-SPSS© ver. 26.0.1 was used for statistical analyses, with significance α-levels estimated to be 5%.

## 5. Results

### 5.1. Study Population Characteristics

Over the five years study period, 8888 TEE exams were retrieved and included in the study. At the time of TEE exams, AF was present in 1959 patients (22%).

Demographic and clinical characteristics of the studied population are shown in Table 1. No differences were detected in patient characteristics between the two center populations, accordingly, data are presented as a whole. The median age (25–75th percentiles) was 71 (65–78) and two-thirds were male and two-thirds were male. A large proportion of the patients presented with a picture of heart failure (HF), and reduced left ventricular ejection fraction (LVEF) ≤40% was found in 46% of the patients. A wide range of cardiac diseases were recognized (see Table 1). Among patients with LAA thrombus, 256 patients (96.6%) presented with NVAF, predominantly persistent or permanent, while nine patients (3.4%) were in sinus rhythm at the time of TTE exam. The median (25th–75th percentiles) CHA2DS2-VASC score was 4 (3–5). Echocardiographic parameters are described in Table 2. Thirty-three patients (12.5%) suffered an embolic event before the echocardiographic diagnosis of the LAA thrombus, representing the reason for the echocardiographic exam, aimed to exclude cardioembolic sources. Among these patients, cardioembolic stroke resulted as the most common thromboembolic presentation (70%), followed by transient ischaemic attack (21%) and acute peripheral artery ischemia (9%).

Due to the retrospective nature of the study, we have no data on peripheral embolization at follow-up.

Among patients with identified LAA thrombus, 183 patients (69%) had a follow-up TEE within one year (mean time to repeat TEE 44 days) and documented continuous anticoagulant therapy, whereas 82 (31%) of the entire population did not undergo a follow-up TEE after the first echocardiographic diagnosis (see Figure 2).

### 5.2. Anticoagulant Treatment

The anticoagulant regimens of patients with LAA thrombus after TEE are described in Table 3. At baseline, 153 (58%) out of 265 patients with atrial/LAA thrombus had been on anticoagulant therapy for at least 3 weeks before the diagnosis—123 patients (80%) were treated with VKA, 10 patients (6.5%) with heparin, and 20 patients (13%) with NOAC.

After diagnosis of LAA thrombus, anticoagulant therapy was started or continued in 258 out of 265 patients (97.4%). VKA was prescribed in 138 patients (52%), heparin in 49 patients (18.5%), and NOAC in 71 patients (27%) (Figure 3). Anticoagulant therapy was contraindicated and not administered in seven patients (2.6%).

### 5.3. LAA Thrombus Resolution Rate

LAA thrombus resolution was assessed in 183 patients who had at least two TEE studies. Among these patients, 122 patients (67%) showed resolution of the LAA thrombus. In contrast, LAA thrombus persistence was seen in 61 patients (33%) at the repeated TEE. Among these patients, NOACs were documented to be ineffective in one; four did not take any anticoagulant therapy due to prohibitive bleeding risk.

In fact, among those treated with VKA who underwent a repeat TEE control, 60 out of 107 patients (66%) demonstrated resolution of the LAA thrombus, Regarding the subgroup of patients treated with LMWH who underwent a second TEE, resolution of LAA thrombus was detected in 21 out of 35 patients (60%), with a thrombus persistence seen in 14 patients (40%). Of the 71 patients prescribed NOACs after the LAA thrombus diagnosis (27.5% of the whole population), 24 patients (9.1%) were prescribed apixaban, 24 (9.1%) dabigatran, 18 (7%) rivaroxaban, and 5 (2.3%) edoxaban. Overall, among subjects assuming NOACs who underwent a repeat TEE, 39 patients (71%) had thrombus resolution, whereas thrombus persistence was seen in 11 treated patients (29%).

In the 122 patients with resolution of the LAA thrombus, TEE showed at follow-up a complete resorption without any sign of fibrosis or consolidation of clots.

Although the rate of thrombus resolution was higher in the subgroup treated with NOACs, the difference between the different anticoagulation regimens was statistically non-significant (VKA, OR 0.9, *p* = 0.83; NOAC, OR 1.23, *p* = 0.42; heparin, OR 0.69, *p* = 0.35).

### 5.4. Predictors of LAA Thrombus Resolution after Anticoagulant Treatment

Patients without thrombus resolution in TEE studies were more likely to be older, male, hypertensive, with higher CHA2DS2-VASc score, and most often presenting with a history of structural heart disease, of which coronary artery disease was the most prevalent. A history of HF, was numerically more frequent in those without thrombus resolution. LAA morphologies were reported in 126 patients (47%) of the examined population. Of these, 26 patients (21%) had the “chicken wing” morphology. Morphologies other than “chicken wing” accounted for 79% (100 patients) of the assessed subgroup. In detail, the prevalence was 42%, 41%, and 17% for the “cauliflower”, “windsock”, and “cactus” morphology, respectively. No significant differences were found in LAA thrombus resolution among different LAA configurations in our study.

As shown in Table 4, upon multivariate-adjusted analysis (Table 4), higher LAA emptying velocity was the only predictor of thrombus resolution, as assessed in 89 patients. Notably, a median value of 20 cm/sec or more was associated with resolution of LAA thrombus (*p* = 0.018). No other clinical or echocardiographic characteristic was predictive of thrombus persistence or resolution at multivariable comparisons in our study population.

## 6. Discussion

The main findings of our study are: (1) the observed prevalence of LAA thrombus in a large unselected cohort of patients undergoing TEE was 3%; (2) among patients with LAA thrombus, only 58% were already on anticoagulant therapy; (3) LAA thrombus resolution on repeat TEE after a period of anticoagulation was seen in 67% of patients; (4) the efficacy of different anticoagulants, including VKA, heparin, and NOACs, on the resolution of LAA thrombus could be comparable; (5) LAA emptying velocity was the only predictor of thrombus resolution.

Our real-world retrospective study showed a 3% prevalence of LAA thrombosis among 8888 TEE exams performed in unselected patients from two high-volume centers over five years. In patients with atrial fibrillation and embolic events, there was a prevalence of LAA thrombus ranging from 8% to 30%Thus, the lower prevalence found in our study may be related to the inclusion in the analysis of all patients who underwent TEE. Differently from other studies, we included non-selected patients undergoing TEE and excluded SEC and sludge from our analysis. Notwithstanding this, our sample population may yield an accurate estimate of the prevalence of LAA thrombosis in unselected patients undergoing TEE. Unexpectedly, nine patients in sinus rhythm were found to have LAA thrombosis.

Our study identified a significant number of patients affected by LAA thrombus despite anticoagulant therapy (58% of the population), which would have been considered, presumably, protected by anticoagulation and suitable for electrical cardioversion. This finding may have a clinical relevance in showing an unacceptable residual thromboembolic risk despite anticoagulation. The retrospective nature of our study did not allow us to assess patient adherence to the anticoagulant regimen. An inadequate anticoagulant therapy, in terms of pill number or dosage, could have influenced the high prevalence of LAA thrombus in a population deemed to be anticoagulated.

No statistically significant difference was found in the rate of LAA thrombus resolution between the different anticoagulant drugs. Although this observation may be limited by the relatively small sample size of patients with LAA thrombus, to our knowledge, our sample population included the largest number of patients with LAA thrombus.

Thus, our data show that, although current ESC guidelines recommend to perform electrical cardioversion without TEE guidance after 3 weeks of therapeutic anticoagulation [1], a relevant residual risk of LAA thrombus may remain despite anticoagulation, and, therefore, the associated risk of cardioembolic events. Consistent with our findings, evidence from the literature show a residual risk of cardioembolic stroke after elective cardioversion, even after the guidelines’ recommended three weeks of anticoagulation, estimated to be ranging between 0.3% and 0.8% [7].

Various echocardiographic parameters have been reported to be predictive of the persistence of LAA thrombus in studies conducted on a selected population [8], namely a lower LVEF, abnormal LAA emptying velocity (<40 cm/s), LAA morphology, diastolic parameters of high LV filling pressure (such as left atrial volume > 34 mL/m^2^ and higher E/e’ mean ratio), or coexistence of mitral valve disease (severe regurgitation or moderate-to-severe stenosis, mitral valve prosthesis). In our population, the only multivariable-adjusted determinant of thrombus resolution was an LAA emptying velocity median value of at least 20 cm/s. In particular, the predictive value of LAA configuration in the risk of developing LAA thrombus with subsequent higher risk of thromboembolic events has not been confirmed in our analysis.

These results emphasize the difficulty of predicting an individual’s risk for thrombus persistence reliably. This may be of greater significance when medication adherence and adequacy of anticoagulation cannot be easily assessed.

Previous studies with warfarin showed thrombus resolution rates ranging from 50% to 90% [9,10]. Although the efficacy of NOACs in the prevention of thromboembolic complications of NVAF has already been tested in previous RCTs [1], few data are available about their efficacy in the resolution of established LAA thrombi, especially in patients undergoing electrical cardioversion or after embolic events.

A sub-analysis of the RELY-AF study suggested that dabigatran could be a reasonable alternative to warfarin in patients requiring cardioversion since the rates of stroke and major bleeding within 30 days of cardioversion on the two doses of dabigatran were low and comparable to those on warfarin with or without TEE guidance [11]. Thereafter, the prevalence of intracardiac thrombi was found to be lower under dabigatran therapy than under vitamin K antagonist in a high-risk patient cohort [12]. Data are still controversial about the efficacy of dabigatran in LAA thrombus resolution. The ongoing prospective, multicenter, randomized controlled clinical trial RE-LATED AF will assess whether dabigatran results in a faster complete LAA thrombus resolution as compared to vitamin K antagonist phenprocoumon [13].

Few data are available about the efficacy of rivaroxaban in patients with AF undergoing electrical cardioversion, appearing, however, to be an effective and safe alternative to VKAs in the X-VeRT randomized clinical trial [14]. In addition, the X-TRA study demonstrated comparable resolution rates of LAA thrombi with rivaroxaban and warfarin, and the CLOT-AF registry reported a 62.5% thrombus resolution with rivaroxaban, suggesting its potential role for the treatment of LAA thrombi [15].

A sub-analysis of the ARISTOTLE trial, as well as data from the EMANATE study [16], demonstrated that the major cardiovascular events after cardioversion of AF, namely thromboembolic events, deaths, and bleedings, are rare and comparable between warfarin/heparin and apixaban [17]. The prevalence of LAA thrombus in NVAF patients treated with apixaban is not well defined, but a study assessed a 3.1%. prevalence among patients undergoing ≥4 weeks of anticoagulation [18]. This suggests that continuous therapy with apixaban does not completely eliminate the risk of LAA thrombus and that TEE prior to cardioversion or catheter ablation may be of benefit in patients with multiple risk factors.

Furthermore, a sub-analysis of the ENGAGE AF TIMI 48 trial and results from the ENSURE- AF trial demonstrated that thromboembolic and major bleeding events post cardioversion were infrequent and similar with edoxaban and warfarin [19], suggesting the potential use of this NOAC in the prevention of cardioembolism in NVAF patients suitable for cardioversion. Concerning the efficacy of edoxaban in the LAA thrombus resolution, the ongoing EDO-SP-01-2015 (A Pilot Study on Edoxaban for the Resolution of Left Atrial Thrombosis in Patients with Non-Valvular Atrial Fibrillation, NCT03489395) will assess the resolution rate in patients with NVAF after a four-week anticoagulant treatment.

Finally, a recent meta-analysis shows that NOACs are as efficacious and safe as warfarin in the treatment of LAA thrombus in patients with NVAF [20].

Real-world estimates would also intend to detect the residual thromboembolic risk associated with performing electrical cardioversion of NVAF without TEE guidance in the NOAC era, and identify high-risk patients in whom TEE should be considered before cardioversion.

Data from our real-world study suggest that NOACs may be at least as effective in LAA thrombus resolution as VKA or LMWH and may better represent the real-world effectiveness of these drugs when non-adherence to anticoagulant therapy plays an important role.

However, the management of patients diagnosed with LAA thrombosis while taking an oral anticoagulant remains a challenge. Data are lacking about the optimal management of AF patients with persistent LAA thrombus despite optimized NOACs or VKA therapy. Farkowski et al. reported in their survey that in patients who previously took a VKA, switching from VKA to a NOAC was the most popular strategy. In the event of a contraindication to NOACs, some participating centers opted for increasing the target time in therapeutic range of international normalized ratio values and for a meticulous assessment of adherence to VKAs. Unfractionated heparin or antiplatelet drugs have rarely been used [21]. In patients already treated with NOAC, suggested strategies include switching from one NOAC to another or no change in treatment [22,23,24]. Indications for the optimal management of patients with atrial fibrillation with persistent LAA thrombus despite NOACs or AVK-optimized therapy is still less established. Miwa et al. showed resolution of warfarin- and dabigatran-resistant thrombus by apixaban [25]. If thrombus was still present after a course of oral anticoagulant therapy, another NOAC change or the addition of an antiplatelet agent or low-molecular weight heparin to current treatment was the strategy adopted.

### Study Limitations and Strengths

Our observational study is limited by its retrospective nature. Underreporting compliance with NOACs or AVKs may not be excluded as we did not perform any analyses of Xa factor or evaluation of the time in therapeutic range, which are beyond the scope of the paper. We included in the analysis a large number of unselected TEEs, therefore, the prevalence of LAA thrombus may have been underestimated compared with pre-cardioversion TEEs. The rate of thrombus resolution may be biased since a significant proportion of patients did not undergo control TEE; furthermore, the comparison between OACs proposed in the results may have been severely biased by unaccounted for confounders, such as patients lost at follow-up.

However, our population represents one of the largest cohorts of LAA thrombosis patients in which comparisons of anticoagulant regimens were performed and corroborates the data on the effect of NOACs in the treatment of LAA thrombus in patients with NVAF.

Additional prospective multicenter studies are needed to confirm these findings and identify predictors of LAA thrombus incidence and resolution/persistence while treated with different anticoagulant regimens.

## 7. Conclusions

Our multicenter, retrospective, and observational study reported a 3% prevalence of LAA thrombus in a large TEE registry. Almost half of the whole population affected by LAA thrombosis (58%) was on apparently appropriate anticoagulation at the time of the echocardiographic diagnosis and would have been considered suitable for electrical cardioversion according to current ESC guidelines. Even though 97% of the patients were prescribed adequate antithrombotic therapy after the diagnosis, LAA thrombus resolution was seen in approximately two-third of the patients who underwent a follow-up TEE. These real-world data suggest a residual thromboembolic risk in patients undergoing electrical cardioversion after the recommended 3 weeks of anticoagulation. Therefore, clinicians should balance for each patient the risk of performing electrical cardioversion without TEE guidance against the benefit of a TEE-guided selection of the patients, in order to minimize the risk of cardioembolism. Follow-up studies are warranted to more reliably estimate the clinical implications of these findings.

## Figures and Tables

**Figure 1 jcm-11-01520-f001:**
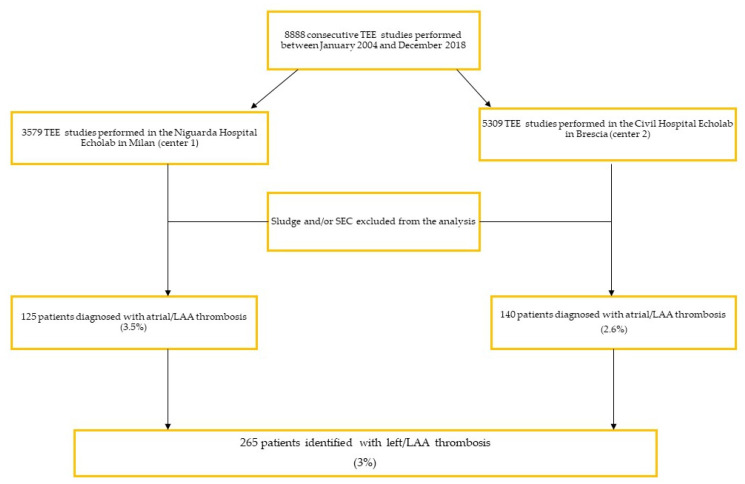
Diagram of the study. EE: transesophageal echocardiogram; SEC: spontaneous echo contrast; LAA: left atrial appendage.

**Figure 2 jcm-11-01520-f002:**
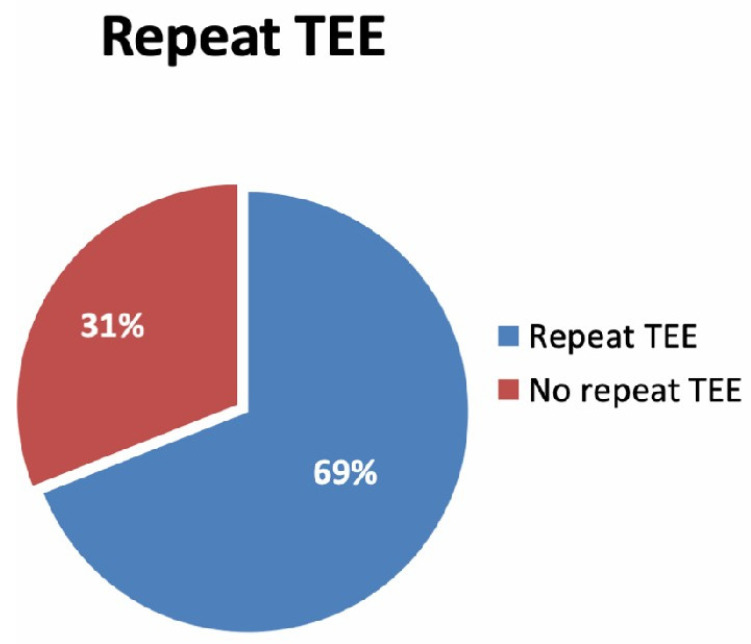
Repeat TTE: TEE, transesophageal echocardiogram. List of abbreviations: TEE, transesophageal echocardiogram.

**Figure 3 jcm-11-01520-f003:**
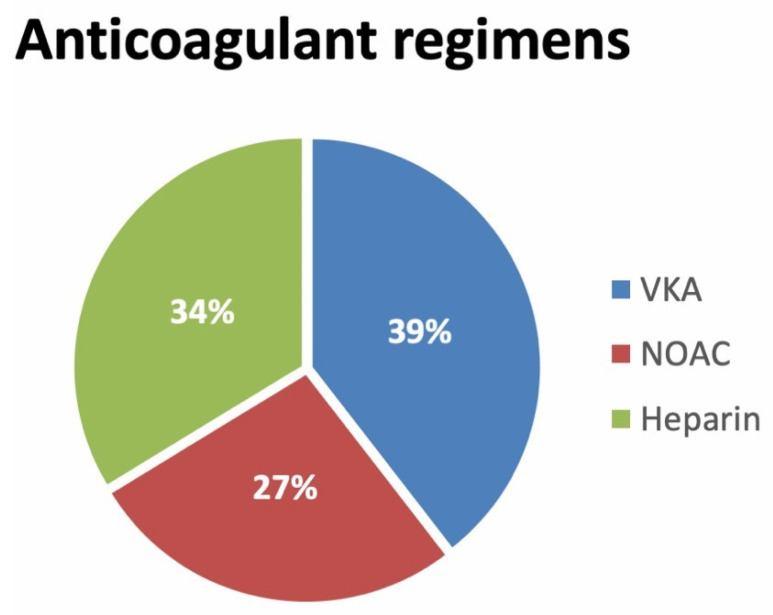
Anticoagulant regimens. List of abbreviations: VKA, vitamin K antagonists; NOAC, non-vitamin K antagonist oral anticoagulants.

**Table 1 jcm-11-01520-t001:** Population characteristics at first TEE. CM, cardiomyopathy; COPD, chronic obstructive pulmonary disease; HF, heart failure; HFmrEF, heart failure with mid-range ejection fraction; HFpEF, heart failure with preserved ejection fraction; HFrEF, heart failure with reduced ejection fraction; TIA, transient ischemic attack.

Characteristic	Entire Cohort
Number of patients	265
Age, years—median (25–75th)	71 (65–78)
Male, n (%)	175 (66)
Atrial Fibrillation, n (%)	256 (97)
Paroxysmal	10 (4.0)
Persistent	163 (64.0)
Permanent	83 (32.0)
Sinus rhythm, n (%)	9(4)
Smoking, n (%)	44(17)
COPD, n (%)	46 (17.4)
Active cancer, n (%)	27 (10)
Diabetes, n (%)	67 (25.3)
Arterial Hypertension, n (%)	204 (77)
HF, n (%)	190 (72)
HFpEF	42 (22.1)
HFmrEF	39 (20.5)
HFrEF	107 (56.3)
CHA2DS2VASc score—median (25–75th)	4(3-5)
History of cardiac disease, n (%)	208 (78.5)
Chronic Coronary Syndromes, n (%)	79 (38)
Valvular, n (%)	53 (26)
Hypertrophic CM, n (%)	5 (3)
Dilated CM, n (%)	26 (13)
Tachycardiomyopathy, n (%)	25 (12)

**Table 2 jcm-11-01520-t002:** Echocardiographic characteristics at first TEE. AP, anterior-posterior; LAA, left atrial appendage; MV, mitral valve.

Echocardiographic Parameters	Entire Cohort (265)
Left atrial dimensions (median) (25–75th)	
AP (mm)	53 (46–61)
Area (cm^2^)	30 (26–36.7)
Volume (mL)	110 (86–152)
E/e’	12 (9–16)
LAA morphology	
Chicken wing, n (%)	26 (21)
Other, n (%)	100 (79)
Cactus, n (%)	17 (13)
Cauliflower, n (%)	42 (33.3)
Windsock, n (%)	41 (32.5)
LAA emptying velocity (cm/sec)-(median) (25–75th)	20 (15–20)
LVEF (%)-(median) (25–75th)	43 (30–55)
>50%, n (%)	85 (41)
40–49%, n (%)	27 (13)
<40%, n (%)	93 (46)
Moderate or severe mitral regurgitation n, (%)	75 (28)
Moderate or severe mitral stenosis n, (%)	13 (5)
Mitral valve replacement n, (%)	33 (12.5)
1. Mechanical valve	26 (79)
2. Tissue valve	71 (21.2)
3. Transcatheter MV intervention (Mitraclip)	4 (1.5)

**Table 3 jcm-11-01520-t003:** Anticoagulant therapy after diagnosis of LAA thrombus. VKA, vitamin K s; NOAC, non-vitamin K antagonist oral anticoagulants.

	N (%)
anticoagulated	258 (97.4)
VKA	158 (52.1)
NOAC	71 (27)
RIVAROXABAN	18 (7)
APIXABAN	24 (9.1)
DABIGATRAN	24 (9.1)
EDOXABAN	5 (1.9%)
HEPARIN	49 (18.5)
Non-anticoagulated	7 (2.6)

**Table 4 jcm-11-01520-t004:** Variables entered on logistic regression: age, gender, CHADS2-VASC2, left atrium diameter (AP= antero-posterior), left atrium volume, EF % (ejection fraction), Severe MR (MR = mitral regurgitation), LAA emptying velocity (LAA= left atrial appendage).

Variables in the Equation with Y = Resolution							
	B	S.E.	Wald	Sig. (p)	OR	95% C.I for Lower	EXP (B) Upper
Age	−0.047	0.060	0.617	0.432	0.954	0.848	1.073
Gender	1.041	1.151	0.818	0.366	2.832	0.297	27.001
CHADS2-VASC2	0.839	0.429	3.828	0.050	2.315	0.999	5.367
Left atrium diameter (AP)	0.020	0.030	0.421	0.517	1.020	0.961	1.082
Left atrium volume	0.012	0.012	1.125	0.289	1.013	0.990	1.036
EF %	−0.050	0.046	1.214	0.270	0.951	0.0870	1.040
Severe MR	0.132	1.009	0.017	0.896	1.141	0.158	8.249
LAA emptying velocity	0.167	0.069	5.789	0.016	1.182	1.031	1.354
Constant	−5.410	5.098	1.126	0.289	0.004		

## Data Availability

All the data underlying this article are available in the article.

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
