# Peer review of "Prevalence and Rate of Resolution of Left Atrial Thrombus in Patients with Non-Valvular Atrial Fibrillation: A Two-Center Retrospective Real-World Study"

_jcm, 2022, doi:10.3390/jcm11061520_

Round 1
Reviewer 1 Report
No further comments
Author Response
We thank the reviewer for the comment
Reviewer 2 Report
The study is important and includes a large cohort of patients with LAA thrombus (265 patients). The authors have nicely shown serveral important issues: 1) Most LAA thrombi develop in patients with persistent or long-persistent AF, (2) in patients with SHD or heart failure LAA thrombi may occur despite antiocoagulant therapy, (3) LAA-appendage velocity is the only predictor for resolution of LAA thrombus within 44 days under continued anticoagulant therapy.
Please add to the discussion how patients without thrombus resolution should be treated: Discuss and cite literature on change of anticoagulant regimen, higher INR-target values (3.0 to 3.5) or OAC plus additional antiplatelet therapy ?
Author Response
The Reviewer raised a good point.
We added this information in the discussion section, as suggested:
“Data are lacking about the optimal management of AF patients with persistent LAA thrombus despite optimized NOACs or VKA therapy. Farkowski et al. reported in their survey that in patients who previously took a VKA, switching from VKA to a NOAC was the most popular strategy. In the event of a contraindication to NOACs, some participating centers opted for increasing the target time in therapeutic range of international normalized ratio values and for a meticulous assessment of adherence to VKAs. Unfractionated heparin or antiplatelet drugs have rarely been used. In patients already treated with NOAC, suggested strategies include switching from one NOAC to another or no change in treatment. Indications for optimal management of patients with atrial fibrillation with persistent LAA thrombus despite NOACs or AVK-optimized therapy is still less established. Miwa et al. showed resolution of warfarin- and dabigatran-resistant thrombus by apixaban. If thrombus was still present after a course of oral anticoagulant therapy, another NOAC change or addition of an antiplatelet agent or low-molecular weight heparin to current treatment was the strategy adopted”.
Reviewer 3 Report
How many of the subjects had evidence of peripheral embolization?
Did the thrombi disappear completely or gradually? Was there any sign of fibrosis or consolidation of clots rather than complete resorption?
Inter- and intra-observor repeatability in detection and characterization of the thrombi?
Author Response

(The authors gave the same response as above.)

Reviewer 4 Report
I read with interest the paper by Faggiano et al. titled ”Prevalence and rate of resolution of left atrial thrombus in patients with non-valvular atrial fibrillation: a two-center retrospective real-world study”.
I thank the authors and the editor for the opportunity to review this interesting manuscript.
I have some reservations as detailed below.
Major points
In this retrospective study, the authors evaluated the prevalence of atrial appendage thrombus in 8888 TEEs in an unselected cohort. Atrial thrombus was observed in 265 or 3% of the patients. Almost all of these patients had AF and slightly more than half of these patients were already treated with an anticoagulant. The TEE was repeated for 69% of the patients of which 67% had thrombus resolution and the authors report that the rate of thrombus resolution was similar for VKA, NOACs, and heparin.
Due to the nature of the study, causality can not be demonstrated whereby the authors should soften their wording e.g. when they state that NOACs have similar efficacy to that of VKA.
Also, there may be differences between the different NOACs which was not evaluated in the study.
The rate of thrombus resolution reported in the paper is limited by the fact that many patients did not undergo repeat TEE or had the repeat TEE done much later. Also the authors do not report the compliance or TTI for VKA before the first and repeat TEE.
At the time of atrial thrombus diagnosis, most of the patients who were already on anticoagulants were treated with VKA. Yet, the authors do not supply data on the INR of the preceding weeks, whereby thrombus formation may relate to dysregulated VKA treatment. Hence, it would not be indicated to do electrical cardioversion without TEE first in some of these patients.
There are some issues with the presentation of data, e.g. the number of patients who were screened, improper reporting of numbers, etc.
The indication for TEE is not reported for both the 8888 evaluations and the 265 patients with thrombus. This would help interpret the results.
There are some grammatical issues. I have commented on some of them below. Consider to use a spell checker or use e.g. Grammarly.
See specific points below.
Abstract
“the efficacy of different anticoagulants” – this study can not evaluate the efficacy of different anticoagulants, rather the authors may report the percentages of thrombus resolution, but the study can not make any causal claims as to the efficacy of different anticoagulants.
“Although the rate of thrombus resolution was higher in patients treated with NOACs” – please add the percentages. The authors should not state that the rate was higher when the rates were not statistically different.
Please use the same number of decimals (zero or one) for all percentages and please revise all the data for the number of decimals throughout the manuscript.
”mean time of 44±66 days” – as this value is not normally distributed, the authors should present median (IQR) and preferably also range.
“the prevalence of LAA thrombus was lower than in selected population” – unclear.
“NOACs were at least as effective as other anticoagulants” - A retrospective can not demonstrate causality and may be confounded by various factors. The authors should soften the language, e.g. that “NOACs could be at least as effective” or similar wording.
The wording of the section “Whats new” should also be revised.
Introduction
“and this percentage results to be higher if the patients undergo a prolonged in-hospital heart rhythm monitoring” – please revise, e.g. that “prolonged in-hospital heart rhythm monitoring allows for a higher rate of AF detection” or similar wording.
“even though patient adherence to this anticoagulant approach remains undetermined” – I would assume that compliance to NOACs have been evaluated in at least several publications. Please supply relevant references.
“particularly, the rate of resolution after the recommended period of anticoagulation therapy.” – some of the patients underwent the repeat TEE more than 100 days after the first evaluation – can the authors tell whether the patients were they still taking anticoagulation?
“3. the rate of LAA thrombus resolution on repeat TEE after a period of anticoagulation“ – please state whether this was after at least three weeks.
Methods
Fig 1. Please define the abbreviations in the figure legend. Also, please state how many patients were excluded. This is a result and should not be referred to in the methods section.
Also, please revise the abbreviations throughout the manuscript, in some instances the authors state e.g. VKA and elsewhere vitamin K antagonists, similar for LVEF and other abbreviations.
“Despite evidence of an association between sludge and thromboembolism5, we did not assess LAA sludge” – was contrast used routinely when in doubt?
What were the inclusion and exclusion criteria? Were patients with valvular AF excluded?
How were the TEEs evaluated? By the clinician who performed the study or by e.g. two experts who reviewed all the evaluations?
Statistical analysis
”Mann-Whitney test” – For data that are normally distributed why do the authors use non-parametric statistics?
Results
Please report how many patients underwent several TEEs, i.e. how many patients underwent the 8888 TEE? Are the repeat TEEs for patients with an atrial thrombus counted among the 8888 evaluations?
“characteristics of the studied population are shown in Table No” – please revise.
“No differences were detected in patient characteristics between the two center populations” – it seems that the prevalence of “History of cardiac disease” was different between the centers. Consider to add * in Table 1 to indicate statistical differences. I agree with the authors that the two centers should be presented as a whole.
“A large proportion of patients had a clinical picture of heart failure (HF), due, in most than half to reduced left ventricular ejection fraction (LVEF)” – Please revise, e.g.: “A large proportion of the patients presented with a picture of heart failure (HF), and reduced left ventricular ejection fraction (LVEF) ≤ 40% was found in 46% of the patients” or similar wording.
How many patients presented with a clinical picture of heart failure?
Less than half (46%) of the patients had reduced LVEF.
“while 9 patients (3.4%) were in sinus rhythm at the time of TTE exam” – how many of these patients had a prior diagnosis of AF or were subsequently diagnosed with AF?
CHA2DS2-VASC score was 4 (SD 1.8) – this is ordinal data and is not normally distributed whereby mean (SD) can not be used. Again, the authors use no decimal and one decimal for the same value in the same sentence.
“Among patients with LAA thrombus, 256 patients (96.6%) presented with NVAF,” – The authors should first report how many patients had a LAA thrombus.
Table 1. Number of male or female patients can be omitted.
Amiloydosis – Amyloidosis.
”Cardioembolic events” – please make this a sub header with stroke, TIA under it to indicate that the 70.0% is of the 33 patients with Cardioembolic events.
Please report the indication for the TEE in a table.
Table 2. Please write the mean (SD), median (IQR), or count (%) as applicable.
5% of the patients had moderate to severe mitral stenosis, yet the authors state that the manuscript deals with non-valvular AF?
Mechanical valve 26 (79%). Do the authors mean Mechanical mitral valve? Is the 79% the percentage of the 12.5% of the patients who had had Mitral valve replacement? If so please place mechanical, tissue valve etc. below a subheader of Mitral valve replacement.
I suppose that Table 1 and 2 refer to the first TEE and not the repeat TEE. Please clarify this point.
The repeat TEE figure can be expressed in a single sentence in the main text and thus, the figure should be omitted. Similar for the Anticoagulant regimens figure.
Please report how long time elapsed between the initial and repeat TEE with median IQR and range.
“were on anticoagulant therapy for at least 3 weeks” – “had been on anticoagulant therapy for at least 3 weeks”.
80% of the patients with atrial thrombus who were already on anticoagulation therapy were treated with VKA and only 13% with NOACs. It would be interesting to know more about the patients with AF who did not have an atrial thrombus, particularly if a larger proportion were on NOACs. The finding of atrial thrombus in patients treated with VKA may relate to dysregulated INR. The manuscript would be improved by the addition of this data and considerations on this subject in the discussion.
“Table 3. Anticoagulant therapy in patients with LAA thrombus” – For clarity, please revise to e.g. “Anticoagulant therapy after diagnosis of LAA thrombus”.
The exact numbers and percentages for the different anticoagulants before and after detection of atrial thrombus should all be placed in Table 3. In the main text the results should be described but all the numbers should not be reported in both the main text and the table.
“Although the rate of thrombus resolution was higher in the subgroup treated with NOACs” – please report first the numbers of thrombus resolution in the different groups.
”VKA, OR 0.9, p=0.83” – Please state what exactly the three P-values refer to.
What do the odds ratio refer to? OR 0.9 for VKA and OR 0.69 appear more different than the rate of thrombus resolution of 66% and 60%, respectively.
“whereas thrombus persistence was documented in 47 patients (34%)” – this can be omitted. These data could be placed in a table for clarity.
“Of the 72 patients prescribed NOACs after the LAA thrombus diagnosis” – In Table 3 only 71 patients were prescribed NOACs. Please revise the data throughout.
“27,5% of the whole population), 24 patients (9.1%)” – again these numbers are presented in a table and should not be repeated here.
“higher value of CHA2DS2-VASc score” – delete “value of”.
“being the coronary artery disease the most represented” – “of which coronary artery disease was the most prevalent”.
”the HF phenotype with” – can be omitted.
“LAA configurations in our real-world study” – delete “in our real-world study”.
Discussion
“The main findings of our study are: 1. the observed prevalence of LAA thrombus in a large unselected cohort of patients undergoing TEE was 3%” – The manuscript would be improved if more information was added about the patients without atrial thrombus. How many patients without atrial thrombus had AF?
“only 58% was already on anticoagulant therapy” – Please revise, do the author mean to state that the patients were not started on anticoagulants or that they had not been on anticoagulation for three weeks?
“was” should be “were”.
“efficacy of different anticoagulants, including VKA, heparin, and NOACs, on the resolution of LAA thrombus was comparable” – again the authors should soften the wording as the study is not designed to evaluate the efficacy of the different ACs.
“Data from current literature reported a higher prevalence of LAA thrombus, ranging from 8% to 30%, in patients with atrial fibrillation and embolic events” – this is a different group from that of the unselected patients in the present study, thus, please delete the word “higher”. Also, “Data from current literature reported” should be “In patients with atrial fibrillation and embolic events a prevalence of … is reported.” or similar wording.
“Unexpectedly, nine patients in sinus rhythm were found to have LAA thrombosis.” – again please report if these patients had a history of AF or were diagnosed with AF subsequently.
“Our study identified a significant number of patients affected by LAA thrombus despite anticoagulant therapy (58% of the population), which would have been considered, presumably, protected by anticoagulation and suitable for electrical cardioversion” – This sentence may be misleading; in the unselected patients only 3% had an atrial thrombus and thus about 1.5% had an atrial thrombus despite treatment with ACs for at least 3 weeks. This may not be representative for patients with newly diagnosed AF as the present data also may include patients with other significant comorbidities, cancer, etc.
It would be very interesting if the authors could supply information about patient compliance to AC for these patients, e.g. INR for the last weeks, number of missed doses of NOAC etc. This information is likely accessible in the charts of most of the patients.
“Thus, our data show that, although current ESC guidelines recommend to perform electrical cardioversion without TEE guidance after 3 weeks of therapeutic anticoagulation1, a relevant residual risk of LAA thrombus may remain despite anticoagulation” – This may be so, but it would be helpful to know how many of the patients who underwent the 8888 TEEs underwent TEE for this indication.
Conclusion
“reported a 3% 5-years- prevalence of LAA thrombus in a large TEE registry” – the same patients were not followed over 5 years. Please revise.
”would be considered” – ”would have been considered”.
Author Response
our answers are attached
This manuscript is a resubmission of an earlier submission. The following is a list of the peer review reports and author responses from that submission.
Round 1
Reviewer 1 Report
- In the statistical analysis section, it is stated that continuous variables are presented as mean and standard deviation. This does not seem to be the case for variables such as left atrial dimensions and LAA emptying velocity in Table 2.
- Figure 2 can be removed as it adds nothing beyond the values already stated in the text.
- Approximately 60% of patients with LAA thrombus were on anticoagulant therapy prior to diagnosis. It would be of interest to know whether any adjustments in dose or type of anticoagulant were made as a result of the diagnosis.
- The authors should be consistent in their use of ‘heparin’ and ‘ low molecular weight heparin’ throughout the text.
- In Table 3, the number of patients receiving rivaroxaban, apixaban, dabigatran and edoxaban doesn’t add up to the total NOAC number.
- The percentages in Figure 3 don’t match those given in the preceding text. The figure can be removed as it is redundant with the text.
- The last paragraph of the results section describes a multivariate analysis (Table 4). Table 4 has not been included with the manuscript. It is stated that a LAA emptying velocity of 20 cm/sec or more as assessed in 89 patients was associated with resolution of thrombus. This is the same LAA emptying velocity (not clear whether it is mean or median) for the whole cohort in Table 2.
- There are a handful of typographical errors to be corrected – Antonella5T5T0052, echocar5diograms,…
Author Response
Reviewer 1:
- In the statistical analysis section, it is stated that continuous variables are presented as mean and standard deviation. This does not seem to be the case for variables such as left atrial dimensions and LAA emptying velocity in Table 2.
Reply: Thank you for this comment. We have corrected and updated Table 2 accordingly.
- Figure 2 can be removed as it adds nothing beyond the values already stated in the text.
Reply: Thank you for this advice. We agree with the Reviewer that Figure 2 may be removed for conciseness reasons. Accordingly, we have deleted this figure and updated figure legends.
- Approximately 60% of patients with LAA thrombus were on anticoagulant therapy prior to diagnosis. It would be of interest to know whether any adjustments in dose or type of anticoagulant were made as a result of the diagnosis.
Reply: Thank you for these insightful comment. According to common practice, type or dose of anticoagulant was changed after the diagnosis of LAA thrombus, unless contraindicated. We already stated in the text the type of anticoagulant regimen used (line 157): “After diagnosis of LAA thrombus, anticoagulant therapy was started or continued in 258 out of 265 patients (97,4%). VKA was prescribed in 138 patients (52 %), LMWH in 49 patients (18,5%), and NOAC in 71 patients (27%) (Figure 2). Anticoagulant therapy was contraindicated and not administered in seven patients (2.6%).” Even if of interest, We think that a deep analysis of the type of adjustment of anticoagulant therapy is beyond the scope of our paper.
- The authors should be consistent in their use of ‘heparin’ and ‘ low molecular weight heparin’ throughout the text.
Reply: Thank you for this suggestion. We have thoroughly checked and corrected the text as suggested.
- In Table 3, the number of patients receiving rivaroxaban, apixaban, dabigatran and edoxaban doesn’t add up to the total NOAC number.
Reply: We apologize for the error and we thank the Reviewer for mentioning that. We have reviewed the data and corrected Table 3.
- The percentages in Figure 3 don’t match those given in the preceding text. The figure can be removed as it is redundant with the text.
Reply: Figure 3 has been deleted
- The last paragraph of the results section describes a multivariate analysis (Table 4). Table 4 has not been included with the manuscript. It is stated that a LAA emptying velocity of 20 cm/sec or more as assessed in 89 patients was associated with resolution of thrombus. This is the same LAA emptying velocity (not clear whether it is mean or median) for the whole cohort in Table 2.
Reply: Table 4 has been deleted from the text
- There are a handful of typographical errors to be corrected – Antonella5T5T0052, echocar5diograms,…
Reply: as suggested, We have thoroughly checked and corrected the text.

Reviewer 2 Report
This is an interesting research addressing
the prevalence of left atrial appendage thrombus
incidentally found on TEE. On top of that, authors
investigated the rate of thrombus resolution after
initiation of anticoagulation. Useful in daily practice
in terms of explaining risk to patients.
Author Response
Reviewer 2:
This is an interesting research addressing the prevalence of left atrial appendage thrombus incidentally found on TEE. On top of that, authors investigated the rate of thrombus resolution after initiation of anticoagulation. Useful in daily practice in terms of explaining risk to patients.
Reply: We gratefully thank the Reviewer for the comments.
Reviewer 3 Report
The authors aimed to investigate 1) the prevalence of LAA thrombus in an unselected cohort of patients undergoing TEE, 2) the efficacy of different anticoagulants, and the determinants of LAA thrombus resolution. Retrospectively 8.888 consecutive TEEs performed over five years in two centers were analyzed. A total of 265 patients (3%) had TEE detected LAA thrombus. After the LAA thrombus diagnosis, VKAs were prescribed in 52%, heparin in 18,5%, and NOAC in 27% of the patients. Among the 183 patients with a repeat TEE (performed at a mean time of 44 days) 67% showed resolution of the LAA thrombus. The authors further found that, NOACs demonstrated to be at least as effective as other anticoagulants in the rate of LAA thrombus resolution, and higher LAA emptying velocities were the only predictor of thrombus resolution.
Comments:
- In the title and throughout the manuscript, “multicenter” should be exactly corrected as “two-centre”.
- It should be clarified, whether all the included patients before the TEEs were diagnosed with AF, i.e. documented ECG with AF?
- Please clarify, 8,888 TEE exams from how many patients?
- Important data are missing: comparison of clinical characteristics between patients with and without LAA Thrombus?
- Comparison of TEE measurements between patients with and without LAA Thrombus?
- Potential predictors for LAA thrombus?
- Besides anticoagulant therapy, was there concomitant antithrombotic therapy?
- Why seven patients were not anticoagulated in view of LAA thrombus?
- Only 69% patients had a repeat TEE, the rate of thrombus resolution can be over-/under-estimated.
- Comparison of the clinical characteristics, the TEE timing and TEE measurements between patients with or without thrombus resolution?
- Clinical thromboembolic and bleeding events should be reported.
Author Response
Reviewer 3:
- In the title and throughout the manuscript, “multicenter” should be exactly corrected as “two-centre”.
Reply: thanks for this advice. The title has been corrected accordingly
- It should be clarified, whether all the included patients before the TEEs were diagnosed with AF, i.e. documented ECG with AF?
Reply: Thanks for this question: AF was present at the time of TEE in 1959 patients (22%)
- Please clarify, 8,888 TEE exams from how many patients?
Reply: Thanks for the question: in the text we add: “performed on 8154 patients
- Important data are missing: comparison of clinical characteristics between patients with and without LAA Thrombus?
Reply: Thanks for this question: in this retrospective study we focused only on patients with LAA thrombus
- Comparison of TEE measurements between patients with and without LAA Thrombus?
Reply: Thanks for this question: in this retrospective study we focused only on patients with LAA thrombus
- Potential predictors for LAA thrombus?
Thanks for this question: please see above
- Besides anticoagulant therapy, was there concomitant antithrombotic therapy?
Thanks for this suggestion: the retrospective nature of this study did not allow us to obtain complete information on other antithrombotic drugs.
- Why seven patients were not anticoagulated in view of LAA thrombus?
Reply: “Seven patients (2.5%) with LAA thrombosis was not prescribed any anticoagulant due to a prohibitive bleeding risk and therefore did not undergo a follow-up TEE”.
- Only 69% patients had a repeat TEE, the rate of thrombus resolution can be over-/under-estimated.
We appreciated this comment and agree
- Comparison of the clinical characteristics, the TEE timing and TEE measurements between patients with or without thrombus resolution?
Thanks for this question: the only finding we found in patients with thrombus resolution was a higher LAA emptying velocity (see the text)
- Clinical thromboembolic and bleeding events should be reported.
Thanks for this relevant point: unfortunately our results are uniquely based on retrospective data from the echocardiographic database ; clinical outcome was not an end-point of our study.
Round 2
Reviewer 1 Report
- In the uploaded manuscript, it appears that Figure 2 was deleted and re-inserted.
- Although 'low molecular weight heparin' was removed from the text, LMWH remains in several places.
- Table 3, and the text below it, states 71 NOAC anticoagulated patients. 18 Riva + 24 Apix +24 Dabi + 6 Edox = 72, unless one patient was treated with multiple drugs.
- As with Figure 2, it appears that Figure 3 was deleted and re-inserted.
- Methods section should describe the multivariate analysis referred to at the end of the Results section.
- 'echocar5diograms' remains in the What's New section.
Author Response
Please see the revised manuscript.

Reviewer 3 Report
As already mentioned in my previous comments, the scientific methodology was the major weakness.
The majority of my questions raised during the first round review are not sufficiently answered at all in the present revised manuscript.
Author Response
Sorry, but i didn't understand what I have to do:
i tried to answer to almost all the request of the reviewers.
I am not able to answer to the request of reviewer 3.
Please see the manuscript.
